# Research on Digital Inclusive Finance Promoting the Integration of Rural Three-Industry

**DOI:** 10.3390/ijerph19063363

**Published:** 2022-03-12

**Authors:** Heping Ge, Bowen Li, Decai Tang, Hao Xu, Valentina Boamah

**Affiliations:** 1School of Management Science and Engineering, Nanjing University of Information Science and Technology, Nanjing 210044, China; 002251@nuist.edu.cn (H.G.); 20201242017@nuist.edu.cn (B.L.); 20215242005@nuist.edu.cn (V.B.); 2Office of Financial Affairs, Jiangsu Open University, Nanjing 210000, China

**Keywords:** digital inclusive finance, integration of rural tertiary industries, financial ecological environment, double-difference method

## Abstract

The development of digital financial inclusion helps create a healthy rural financial ecological environment and plays an important role in integrating rural tertiary industries. This paper incorporates digital financial inclusion into the rural tertiary industry integration research framework. Furthermore, it adopts the double-difference method to empirically analyze the impact of the development of digital financial inclusion on rural tertiary industry integration from the perspective of policy impact. In addition, it considers regional differences in the financial ecological environment; robustness tests were carried out using methods such as placebo tests and validated the conduction mechanism. The study through the double-difference model found that digital financial inclusion is very conducive to promoting rural tertiary industry integration; using the quantile DID (difference in differences) method to analyze the heterogeneity, it is concluded that there is a heterogeneous impact on rural tertiary industry integration. It exerts a more significant improvement in provinces and cities with higher rural tertiary industry integration levels. Constructing an intermediary effect model to verify the transmission mechanism concludes that the policy has promoted the improvement of rural tertiary industry integration efficiency by promoting technological innovation, improving agricultural modernization, and building a risk-sharing mechanism. Finally, it puts forward policy recommendations from optimizing the financial ecological environment, rationally allocating financial resources, and perfecting the transmission mechanism.

## 1. Introduction

How to transform and upgrade traditional agriculture and expand and extend the agricultural value chain are currently the focus of agricultural development in various countries. The integrated development of the three industries in rural areas is an important strategic plan for achieving high-quality economic development, agricultural supply-side reforms, and revitalizing rural industries. China is the largest developing country; the No.1 Central Document in 2021 pointed out that it is necessary to build a modern rural industrial system and empower the integration of rural tertiary industries [1]. Promoting the prosperity of rural industries and enhancing the core competitiveness of rural areas through the integrated development of rural primary, secondary, and tertiary industries is an important part of my country’s cultivation of new momentum for rural revitalization and the path to socialist rural revitalization with Chinese characteristics.

In promoting the development of inclusive finance around the world, more countries have begun to pay attention to the impact of inclusive finance policies on economic and social development, and China is a typical example. For a long time, the central government has always paid full attention to enabling finance to empower rural industries and promote the integration of rural tertiary industries. Thus, from developing policy-based financial institutions to serve rural tertiary industries to reforming how finance assists rural tertiary industries in China, finance and the rural tertiary industries co-exist and prosper together. In order to further solve the financial difficulties faced by the integration of rural tertiary industries, the “Strategic Plan for Rural Revitalization (2018–2022)” clearly pointed out that in the new era to continue to promote the integration of rural tertiary industries, higher-quality funds and more diverse financial services are required [2]. Product types and more diversified financial service methods must build a financial support model with Chinese characteristics. Therefore, the Fourth Plenary Session of the 19th Central Committee of the Communist Party of China officially regarded data as a new factor of production and vigorously developed digital inclusive finance in rural tertiary industries [3]. However, at present, China’s digital inclusive finance to support rural tertiary industry integration is still in its infancy. In the acceleration stage, the vulnerability of the rural financial ecological environment is still relatively prominent, and many problems and challenges that arise cannot be ignored. Therefore, the “14th Five-Year Plan Outline” released in 2021 continues to point out that the development of digital inclusive finance will promote the integrated development of the primary, secondary, and tertiary industries in rural areas [4].

In the “14th Five-Year Plan Outline”, the reason why the central government chose digital financial inclusion as an important way to integrate the tertiary industries in rural areas was that they believed that digital financial inclusion could promote the integration of tertiary industries in rural areas. However, can it be tested in practice? What transmission mechanisms does digital financial inclusion use to promote the integration of rural tertiary industries? How can further efficiency improvement be promoted to better serve the integration of rural tertiary industries? Research on these issues, from the perspective of policy choice, promoting the inclination of financial resources to the rural areas and increasing the credit supply to the tertiary industry is the proper meaning of realizing the integration of the three industries in rural areas. Digital inclusive finance at the time of prosperity provides these new ideas. From an empirical point of view, existing studies have paid more attention to the correlation between digital financial inclusion and rural economy and industrial structure and concluded that digital financial inclusion plays a positive role in all aspects. Therefore, the research in this paper is reasonable and conducive to accelerating the construction of China’s digital inclusive finance and improving the level of integration of the tertiary industries in rural areas. It also provides more possibilities for the integrated development of rural industries in various countries.

Given this, the main objectives of this paper are to fully understand the impact of the development of digital financial inclusion on the integration of rural tertiary industries, and from the perspective of policy impact, taking China as an example, using the double-difference model for regression analysis. On this basis, it further analyzes the specific effects of the integration of rural tertiary industries under the adjustment of the financial ecological environment. This study expands the research perspective of the environmental factors of digital financial inclusion influencing the integration of rural tertiary industries. A series of robustness tests are carried out to verify the reasonableness of the results. With the help of the quantile DID (difference in differences) method, a heterogeneous analysis is carried out, revealing that digital financial inclusion has a heterogeneous impact. Furthermore, the construction of the mediation effect model verified the three transmission mechanisms, making the results more convincing.

The possible contributions of this paper are mainly reflected in the following: first, selecting the “G20 Advanced Principles of Digital Financial Inclusion” as a quasi-natural experiment, using the double-difference method to conclude that the development of digital financial inclusion is significantly conducive to promoting the integration of the three industries in rural areas, and considering the specific effect of the financial ecological environment on the integration of the three industries in rural areas, this conclusion still holds after a series of robustness tests. Second, using the quantile DID method to analyze the heterogeneity, it is concluded that the promotion effect on the provinces and cities with a higher level of rural integration of the three industries is more significant. Third, this paper tries to examine the transmission mechanism of digital financial development to support the integration of the three industries in rural areas. Based on the verification of the intermediary effect model, it is concluded that there are three channels for digital inclusive finance to support the integration of the three industries in rural areas: technological innovation, agricultural modernization, and risk-sharing. It is expected to provide a theoretical basis and decision-making reference for promoting digital inclusive finance to support the integrated development of rural tertiary industries on a global scale. In addition, this paper aims to provide enlightenment for other countries to promote integration of rural tertiary industries.

## 2. Review of Literature

The implementation of the rural revitalization strategy must focus on integrating the primary, secondary, and tertiary industries in rural areas and scientifically plan the development of the industry. Since the integration of the three industries in rural areas was proposed, it has quickly become a hot issue in society. Barbara Braun et al. believed that the essence of integrating primary, secondary, and tertiary industries in rural areas is to integrate different industries into agriculture to promote sustainable development of agriculture, environment, and economy [5].

Some discussions on the integration of the three industries in rural areas are from industrial chain integration. Knutson, R.D. pointed out that industrial integration usually has to go through a three-stage process of technology running-in, product and service business integration, customer market integration, and finally takes the consumer market as the guidance [6]. Only when the agricultural industry chain is connected with information technology and through reverse integration can it adapt to continuous upgrading and changes [7]. Some scholars have broadened their horizons to the mechanism of linking interests among various subjects, which can positively impact small farm enterprises by developing and implementing models that meet basic needs and open new markets and production plans [8]. Other scholars use science and technology as the mainline of exploration, pointing out that the deep integration of science and technology can promote the birth of new industries and expand existing industries, thereby realizing industrial integration [9]. Existing results show that the financing support provided by financial institutions is conducive to the development of rural industries. However, the profit-seeking nature of capital makes many financial resources flock to urban industries, resulting in a general shortage of rural financial resources and a relative lack of funds for rural tertiary industries [10]. The emergence of digital financial inclusion is conducive to improving the situation.

Digital financial inclusion enables customers who cannot be covered by traditional finance to receive high-quality financial products and services [11,12], focusing on inaccessible and affordable financial services in rural or remote urban areas. Therefore, analyzing the rural economic effects of digital inclusive finance has always been a research hotspot for scholars. One type focuses on the income effect. Grossman and Tarazi found that digital finance can help Kenyan farmers increase their income by facilitating payment and smoothing consumption [13]. Dahiya and Kumar, based on an empirical study of India’s emerging economy, found that digital financial inclusion can reduce rural income inequality [14]. The other category focuses on financial effects. Combining digital technology and inclusive finance is significant for rural low-income groups to obtain equal financial services [15]. This helps to break through the limitations of time and space and allows the poor and disadvantaged groups to obtain financial services on demand while promoting inclusive finance [16,17,18,19,20].

The time when digital inclusive finance supports the integration of the three industries in rural areas is relatively short. Few pieces of literature directly analyze the impact of digital inclusive finance on the integration of the three industries in rural areas. Still, the research results on the impact of finance on the industrial structure are worthy of reference. Finance needs to return to the real economy, and economic financialization will hinder the optimization of industrial structure [21]. Through the role of the digital inclusive financial market, the coverage of financial services can be greatly improved, and capital can be effectively allocated [22,23], thereby optimizing the allocation of funds among various industries and promoting the further upgrading of the industrial structure [24,25,26,27].

Some scholars believe that the discussion on the transmission mechanism of digital financial inclusion should be strengthened. In terms of improving the innovation level of enterprises, digital finance realizes the information exchange between “financial sector-enterprise main body” by mining massive standard and non-standard data to meet the financial needs of enterprises as a transmission channel [28]. In terms of narrowing the urban–rural income gap, digital finance under the background of the Internet revolution stimulates income growth by benefiting groups excluded from traditional finance and easing financing constraints [29]. In terms of stimulating consumption, digital finance mainly promotes household consumption through transmission channels such as improving convenience, improving information transparency, enriching the use of funds, and increasing a sense of security, especially for low-income people living in third- and fourth-tier cities [30].

In terms of the ecological environment of digital financial inclusion, Asongu found that finance and credit, society, economy, and other environmental elements interact and cooperate, giving full play to the role of each environmental element for coordinated development, which is conducive to the formation of a more healthy and orderly financial ecology environment [31]. Some scholars have analyzed the impact of the economic environment on the efficiency of digital financial inclusion. The higher the level of external financial development, the lower the information asymmetry, making it less difficult for enterprises to obtain external funds [32,33]. Some scholars start from the perspective of the financial development environment; Arizala et al. empirically tested that the factor of financial development showed a positive role in promoting total factor productivity at the macro level [34]. Establishing a diversified financial ecosystem can effectively improve the efficiency of financial institutions [35,36]. Other scholars have found from the perspective of the government environment that a good government governance environment will positively impact the operational efficiency of the inclusive financial system [37,38].

The literature review above shows that the existing literature has achieved relatively rich results from the research on the support of the rural economy, industrial structure, transmission mechanism, and rural three-industry integration by digital inclusive finance. These works of literature help us to deepen our research. However, many studies focus on measuring the level of integration of the three industries in rural areas. At present, few scholars have evaluated the level of integration of the three industries in rural areas supported by digital inclusive finance. In addition, most scholars’ studies only analyze the influencing factors and fail to identify the transmission mechanism, which may not effectively make policy recommendations. Therefore, this paper will analyze the impact of digital financial inclusion on the integration of the three industries in rural areas from the perspective of policy impact, discuss the heterogeneity, verify the transmission mechanism, and provide a new perspective of digital inclusive finance for the integration of the three industries in rural areas.

## 3. Theoretical Mechanism and Model

### 3.1. Theoretical Mechanism

This paper analyzes the theoretical mechanism of the influence of digital inclusive finance on the integration of the three rural industries. This paper considers the effect of the policy on the efficiency of rural integration through three transmission channels: technological innovation, agricultural modernization, and risk-sharing.

Digital inclusive finance improves the efficiency of rural tertiary industry integration by rising technological innovation. On the one hand, the interconnection of digital inclusive finance and the improvement of infrastructure can effectively reduce the information asymmetry of the capital market and create a good financial environment for technological innovation. On the other hand, promoting technological innovation capacity is conducive to promoting the integration of rural tertiary industry. The higher the level of financial creation, the higher the efficiency of the financial screening of the funds to be invested. The greater the probability of successful financing and innovation of rural enterprises, the higher the technical level, essentially improving the efficiency of resource utilization and productivity; it plays a vital role in developing the integration of the three rural industries.

Therefore, Hypothesis 1 is put forward: digital inclusive finance improves the efficiency of rural tertiary industry integration by encouraging technological innovation. Digital inclusive finance promotes rural tertiary industry integration efficiency by improving agricultural modernization. On the one hand, optimizing rural financial supply can provide necessary financial support for agricultural modernization and promote agricultural modernization. Digital inclusive finance promotes agricultural transformation and upgrading and uses new technologies to improve industrial efficiency. On the other hand, improving agricultural modernization is conducive to integrating the three rural industries. The higher the level of agricultural modernization, the more effective the inter-and intra-industry cooperation is, which helps to save production cost, improve production efficiency, give full play to scale effect and structural dividend, and support the integration of the three rural industries.

Therefore, Hypothesis 2 is put forward: digital inclusive finance promotes the efficiency growth of the integration of the three industries in rural areas by improving agricultural modernization.

Digital inclusive finance helps realize the integration of rural industry and industry through the risk-sharing mechanism. On the one hand, the development of digital inclusive finance is conducive to constructing a risk-sharing mechanism. First, inclusive finance can significantly promote the development of rural inclusive insurance; second, digital finance can make it easier for consumers to use risk management tools to mitigate concerns about uncertainty in the future. On the other hand, constructing a risk-sharing mechanism is conducive to integrating rural tertiary industry. The current risk-controlled release method allows farmers to diversify management risks through multiple channels, ensure product returns, improve the possibility of farmers entering the rural three-industry integration market, and help rural three-industry integration.

Therefore, Hypothesis 3 is put forward: digital inclusive finance can improve the efficiency of rural integration by constructing a risk-sharing mechanism.

### 3.2. Theoretical Model

Based on China’s digital inclusive finance practice, this paper analyzes the resource allocation behavior of digital inclusive finance and whether it can affect the efficiency of rural tertiary industry integration.

In order to analyze the influence of digital inclusive finance on the efficiency of rural tri-industry integration, this paper uses the methods of Odedokun for reference [39]. Starting from the traditional production function, it takes digital inclusive finance and rural labor force as input factors, and the efficiency function of rural tertiary industry integration is obtained:(1)Y=AN+F(N,C)

Among them, Y represents the efficiency of the rural integration of the tertiary industry; A represents the efficiency of production; N represents the level of financial support of the figure Pratt & Whitney; C represents the input of rural labor force.

F(N,C) is a neoclassical production function that satisfies the scale reward constant,$\underset{N\to 0}{\lim}\frac{\partial F(N,C)}{\partial N}=\infty$, $\underset{N\to \infty }{\lim}\frac{\partial F(N,C)}{\partial N}=0$, $\underset{C\to 0}{\lim}\frac{\partial F(N,C)}{\partial C}=\infty$, $\underset{C\to \infty }{\lim}\frac{\partial F(N,C)}{\partial C}=0$. Form (1) can be converted into per capita expression:(2)y=An+f(n)

Among them, y=Y/C and n=N/C respectively represent the per capita contribution rate and per capita available capital of rural tertiary industry integration under the support of digital inclusive finance. f(n)=F(N,C)/C. With the development of digital inclusive finance, more and more rural savings will make an innovative investment through digital finance channel. More and more funds will be invested in the rural tertiary industry, which will affect the efficiency of rural tertiary industry integration. If the saving rate is h and the investment rate is u, the total support of digital inclusive finance in rural integration is: T1=huY. Assuming that, without depreciation, the change in available capital equals investment and the total population remains the same, $T1=d_{n}$, then the rate of growth of available capital per capita is:(3)$$g=\frac{d_{n}}{n}=\frac{d_{N}}{N}-\frac{d_{C}}{C}=huA+hu\frac{f(n)}{n}$$

From Equation (3), we can find that with the increase of capital brought by digital inclusive finance, the capital available rate per capita g will decrease gradually, which indicates that the growth rate of available rural capital will slow down. According to Equation (2), in the steady-state of the economy (n→∞), we can calculate that the growth rate of the per capita contribution rate of the rural tertiary industry I is:(4)$$\underset{n\to \infty }{\lim}\frac{dy}{y}=\underset{n\to \infty }{\lim}\frac{Ad_{n}+f(n)d_{n}}{An+f(n)}=\frac{d_{n}}{n}=huA+hu\frac{f(n)}{n}$$

Equation (4), by taking the derivative of the number Pratt & Whitney and the degree of innovation of finance u, we get:(5)$$\frac{\partial_{i}}{\partial_{u}}=hA+h\frac{f(n)}{n}>0$$

Based on the above theoretical mechanism, digital inclusive finance has three transmission channels: technological innovation, agricultural modernization, and risk sharing. Based on the ideas of Acemoglu and Pischke [40], this paper constructs an optimal model of human capital investment for rural integration. This article makes the following assumptions:

First, a large number of the rural labor force to technological innovation as a source of income, after the success of income $I_{tech}$, can be obtained, at the same time O will be the output of rural integration.

Secondly, those engaged in integrating the three rural industries need to carry out the agricultural modernization reform according to the real possibility of obtaining higher income. Still, the excessive cost may cause the rural laborers to stop carrying out the agricultural modernization reform, thereby reducing the probability of achieving agricultural modernization. If $I_{\arg}$ a random income shock followed successful selection and agricultural modernization, the cumulative distribution function would be B. At this point, the real profit of the rural labor force is $O-I_{tech}+I_{\arg}$, if $O-I_{tech}+I_{\arg}\geq 0$ and only if the rural workers are willing to continue agricultural modernization reform, otherwise the rural workers withdraw from the three-production integration market. The probability that rural workers will continue the agricultural modernization reform is $I_{\arg}(O,I_{tech})=\mathrm{Pr}(I_{\arg}\geq I_{tech}-O)=1-B(I_{tech}-O)$. Because it is a non-decreasing function, the more advanced the enterprise technology, the higher the cost of using advanced technology and the lower the probability of continuing agricultural modernization reform.

Third, there are risks involved in integrating the rural tertiary industry, which will have a certain degree of utility damage to the rural workers. As a result, rural integration workers have to pay for insurance j and other services to reduce risk. It is further assumed that access to risk-sharing services entails costs and that without the purchase of risk-sharing services $\left(I_{ins}=0\right)$, the level of risk exposure can not be reduced to the greatest extent possible θ, but only F(θ) the cumulative distribution of the level of risk exposure can be known.

Fourth, the benefits of purchasing risk-sharing services come from two aspects: reducing the loss when the risk occurs. The risk mitigation effect is $I_{tech}-J(I_{ins})\theta$. Among them, $J(I_{ins})$ measures the risk-sharing mechanism to reduce the risk negative utility function. If $I_{ins}=0$, $J(I_{ins})=1$, the rural laborer entirely bears the negative utility of the risk; if $I_{ins}>0$, $0<J(I_{ins})<1$ and $J^{\prime }(I_{ins})<0,J^{\prime \prime }(I_{ins})<0$, the more investment in the risk-sharing mechanism, the more the loss can be reduced. Second, through the purchase of risk-sharing services to promote the integration of rural tertiary industry, that is $O=O(I_{ins})$, and $O^{\prime }(I_{ins})>0,O^{\prime \prime }(I_{ins})<0$. Under the assumptions mentioned above, workers engaged in the integration of the three rural productions following the expected market conditions decide whether to continue technological innovation and promote agricultural modernization and risk-sharing:(6)$$\underset{I_{tech,}I_{ins}}{\max}\Delta =\left[1-B(I_{tech}-O)\right]\int_{0}^{\theta }\left[I_{tech}-J(I_{ins})\theta \right]dF(\theta )-I_{ins}$$

Among them, Δ represents the utility of the workers engaged in the integration of the three rural industries, θ>0 represents the risk threshold of labor participation. When the risk of those engaged in the rural integration is higher than θ, they will withdraw from the rural integration because of the high risk. The first-order conditions of the optimization problem are:(7)$$\frac{\partial \Delta }{\partial I_{tech}}=\left[1-B(I_{tech}-O)\right]F(\tilde{\theta })-\frac{\partial B}{\partial I_{tech}}\int_{0}^{\tilde{\theta }}\left[I_{tech}-J(I_{ins})\theta \right]dF(\theta )=0$$
(8)$$\begin{array}{ll} \frac{\partial \Delta }{\partial I_{ins}} & =-\frac{\partial B}{\partial O}\frac{\partial O}{\partial I_{ins}}\int_{0}^{\tilde{\theta }}\left[I_{tech}-J(I_{ins})\theta \right]dF(\theta )-\left[1-B(I_{tech}-O)\right]\int_{0}^{\tilde{\theta }}\left[J^{\prime }(I_{ins})\theta \right]dF(\theta )-1\\ & =0 \end{array}$$

Since $\frac{\partial B}{\partial I_{tech}}=-\frac{\partial B}{\partial O}$, simultaneous Equations (7) and (8) can obtain the optimal level of technological innovation $I_{tech}{}^{*}$, and risk-sharing expenditures $I_{ins}{}^{*}$, they must satisfy:(9)$$\left[1-B\right]\left[O^{\prime }(I_{ins}{}^{*})F(\tilde{\theta })-J^{\prime }(I_{ins}{}^{*})Q(\tilde{\theta })\right]=1$$

Among them, $Q(\tilde{\theta })=\int_{0}^{\tilde{\theta }}dF(\theta )$ represents the average value of the risks faced by the personnel engaged in the rural integration of the tertiary industry, $F(\tilde{\theta })$ indicates the proportion of the rural integration labor force that meets the threshold of participation risk. In general, it can be assumed that the overall workforce $Q(\tilde{\theta })$ and $F(\tilde{\theta })$ is independent of the individual level of technological innovation $I_{tech}{}^{*}$ and that risk-sharing mechanisms $I_{\mathrm{ins}}{}^{*}$ are in place. According to Formula (9), under certain conditions, for example $O^{\prime \prime }(I_{ins}{}^{*})F(\tilde{\theta })-J^{\prime \prime }(I_{ins}{}^{*})Q(\tilde{\theta })>0$, the optimal risk-sharing expenditure $I_{\mathrm{ins}}{}^{*}$, and the optimal level of technological innovation $I_{tech}{}^{*}$, are positively correlated ($dI_{ins}{}^{*}/dI_{tech}{}^{*}>0$); that is, individuals who expect higher levels of technological innovation tend to spend higher risk-sharing expenditure. It is necessary to reduce the negative utility caused by the failure and increase the integration efficiency of the three rural industries through risk-sharing.

The optimization problem Equation (6) does not consider the budget constraints faced by individuals. In Equation (6), the subsidy provided by the government to engage in the rural integration of the three industries v is introduced, and Equation (6) is rewritten as:(10)$$\underset{I_{tech},I_{ins}}{\max}\left\{0,\Delta =\left[1-B(I_{tech}-O)\right]\int_{0}^{\tilde{\theta }}\left[I_{tech}-J(I_{ins})\theta \right]dF(\theta )-I_{ins}+v\right\}$$

In theory, if the government subsidy income can increase the individual’s risk-sharing expenditure by stimulating technological innovation and popularizing insurance and other services, $\Delta \left(I_{tech},I_{ins}\right)\geq 0$ must be established; otherwise, the laborers will withdraw from the rural triple-industry integration market. The choice does not carry on the countryside three production fusion. On behalf of the probability of agricultural modernization $\left[1-B(I_{tech}-O)\right]$, given the budget constraints $\Delta \left(I_{tech},I_{ins}\right)\geq 0$, the probability of agricultural modernization will increase, encouraging workers to increase risk-sharing expenditure and try higher levels of technological innovation. The development of digital inclusive finance involves increasing government subsidies and increasing the rural population’s access to financial support and risk-sharing through the optimization Equation (10).

In addition to the two mechanisms of technological innovation and agricultural modernization, the independent risk-sharing mechanism can also improve the rural integration of the three industries. Under the condition of not considering the level of technical innovation and realizing agricultural modernization, the choice problem that the rural three-industry amalgamation laborer faces becomes:(11)$$\underset{I_{ins}}{\max}\Delta =\int_{0}^{\tilde{\theta }}\left[w-J(I_{ins})\theta \right]dF(\theta )-I_{ins}+v$$

Among them, w is engaged in the rural three-industry integration of workers that can obtain income, at this time budget constraint is $w+v\geq I_{ins}$. The first-order condition is: $\int_{0}^{\tilde{\theta }}\left[-J^{\prime }(I_{ins})\theta \right]dF(\theta )-1=0$. Optimal risk-sharing expenditure level is $J^{\prime }(I_{ins}{}^{*})=-1/Q(\tilde{\theta })$. As we assume $J^{\prime \prime }(I_{ins}{}^{*})<0$, then $-J^{\prime \prime }(I_{ins}{}^{*})>0$ is $-J^{\prime }(I_{ins}{}^{*})$ and therefore an increasing function of $I_{ins}{}^{*}$. Before the introduction of the subsidy policy, v=0. If $I_{ins}{}^{*}\geq w$, households spend the w most on the purchase of risk-sharing services, then the exogenous income growth can certainly raise the level of rural tertiary integration; if $I_{ins}{}^{*}<w$, household income is sufficient to purchase the optimal risk-sharing services; the growth of exogenous income is unlikely to lead to an increase in risk-sharing expenditure.

## 4. Research Design, Index Selection and Data Explanation

### 4.1. Metrological Model Design

In order to test the impact of the development of digital financial inclusion on rural tertiary industry integration, this paper uses the double-difference method to regress the efficiency of rural tertiary industry integration in 31 provinces and cities across the country with the virtual variable of whether to develop digital financial inclusion. Considering the regional differences and time lag in the development of digital financial inclusion, traditional DID is only suitable for evaluating policy effects at a single point in time. This paper constructs a DID model, adding province and time fixed effects to control the difference between some uncontrollable factors between provinces and years. The benchmark double difference model is shown in Equation (12):(12)$$Y_{i,t}=\alpha +\beta Key_{i,t}+\varphi X_{i,t}+\mu_{i}+\gamma_{t}+\varepsilon_{i,t}$$

Among them, $Y_{i,t}$ represents the integration efficiency of rural tertiary industry, $Key_{i,t}$ is the core explanatory variable, β is the effect of Policy Implementation, $X_{i,t}$ is the control variable, $\mu_{i}$ is the province fixed effect, $\gamma_{t}$ is the year fixed effect, and $\varepsilon_{i,t}$ is the random interference item.

### 4.2. Variable Description

Explanatory variable: The efficiency of rural integration of tertiary industry. The DEA-Malmquist index method was used to measure rural tertiary industry integration efficiency in 31 provinces and cities in China from 2011 to 2018. Specific output and input indicators are selected as follows: (1) factor input: capital and labor are needed to promote the development of rural integration of the tertiary industry, and in the rural area it is often the government’s first input and other capital inflow. So, the expenditure of agriculture, forestry, and water affairs of 31 provinces and cities in China from 2011 to 2018 was taken as the input index of rural industry integration. This paper uses the number of rural non-agricultural employment to measure the input of the labor force. (2) output indicators: first, economic indicators select the total output value of agriculture, forestry, animal husbandry, and fishery services. Second, social indicators select rural per capita consumption expenditure. Third, the ecological indicators select the level of mechanised agriculture.Core explanatory variable: Whether to develop digital financial inclusion in the virtual variable Treat distinguishes the Experimental Group from the Control Group. If Treat = 1, it means the experimental province in the East; if Treat = 0, it means the comparative province in the Midwest. Time is a virtual variable of policy, and the G20 digital Pratt & Whitney senior principles of finance, published in 2016, formally introduced digital inclusive finance. Therefore, this paper selects 2016 as the policy shock event. The Time value of 2011–2015 is 0 and of 2016–2018 is 1. The interaction item Treat Time explains the variables for the important core.Financial ecological environment variables: The financial ecological environment of a region is closely related to the development of digital inclusive finance, and the development of the financial ecological environment will affect the effect of policy implementation. According to the existing research on the definition of financial ecological environment, the impact on the level of rural tertiary industry integration can be understood from the following three aspects: (1) government governance (Gs), measured by the ratio of fiscal expenditure to GDP; (2) financial development (Fin) is measured by the ratio of the financial loan balance of each province and city to the regional GDP; (3) economic base (Rgdp) is measured by per capita GDP.Control variables: (1) human capital level (Edu), which is measured by the number of students in ordinary colleges and universities in rural areas as a proportion of the total rural population; (2) information level (Info), which uses rural transportation and communication expenditures as a proportion of rural consumption expenditures measured by proportion.Intermediate variable: (1) technological innovation (Tech). Using patent licensing as a metric. (2) agricultural modernization (Arg). Use electricity consumption per square meter of agricultural land. (3) risk diversification (Insurance). Use the value of premium income as a metric.

### 4.3. Data Interpretation

Data selection: The data time span selected in this paper is from 2011 to 2018, covering panel data of 31 provinces and cities in China because China’s digital inclusive finance has grown from birth to maturity since 2011. Since 2015 it has officially become the focus of government policies to promote the integration of the three industries in rural areas, so it is more reasonable to study the data during this period.Data source: Considering the problems such as the measurability and availability of data, the lag of data release, etc. The data are drawn from regional yearbooks and government websites, the 2011–2018 China Statistical Yearbook, and the report on developing China’s agricultural product processing industry. The descriptive statistics for each variable are shown in Table 1.

## 5. Empirical Results and Robustness Test

### 5.1. Baseline Regression Result

In order to test the impact of the development of digital financial inclusion on the integration of rural tertiary industries, this study uses the DID method to perform regression. The benchmark regression results are shown in Table 2. As shown in Table 2, the first column is the regression result without financial ecological environment variables and control variables. It can be seen that the development of digital financial inclusion is significantly conducive to the improvement of the level of integration of the rural tertiary industries. The second column is the regression result of further adding financial ecological environment variables and control variables to control each province’s economic characteristics. It can be seen that the coefficient of the core explanatory variable is 0.2472517. It is significant at the 1% confidence level, indicating that digital financial inclusion positively affects rural tertiary industry integration. The implementation of this policy is significantly conducive to rural tertiary industry integration promotion. At the same time, compared with the test results of the first column of core explanatory variables, Model 2 is better than Model 1, indicating that the financial ecological environment has a positive effect on the integration of rural tertiary industries.

Next, this paper discusses the other explanatory variables of the second column of benchmark regression results. The role of government governance in promoting the integration of the tertiary industry in rural areas is not significant. This reflects the fact that the government may be pursuing political achievements, leading to inadequate allocation of fiscal funds, imperfect supervision of the use of funds, lack of overall planning, and scientific investment decision making. The efficiency of the use of funds is greatly reduced.

The coefficient of financial development is significantly negative. The reason for this may be that vigorously developing the economy will enable financial resources to serve economic growth. At this time, regional institutions are in a profit-seeking mentality, and they do not “take funds from agriculture and use them for agriculture” but use funds instead. Investment in the non-agricultural sector has caused a large outflow of funds, leading to inefficiency.

The economic foundation has a positive and significant impact on the integration of the rural tertiary industries. It shows that strengthening the economic foundation can improve the level of integration of the rural tertiary industries. This is because a high economic level and a reasonable economic structure are the basis for generating economic agglomeration effects, which will bring economies of scale and promote the effective operation of digital inclusive financial capital and the realization of the integration of rural tertiary industries.

The informatization level has an insignificant positive impact on the integration of the tertiary industries in rural areas, indicating that the promotion of the informatization level on the integration of the tertiary industries in rural areas is not obvious enough. This may be related to China’s rural tertiary industry integration still being in the transitional stage of transformation. At this time, the rural tertiary industry integration cannot fully use advanced technologies brought about by informatization. The penetration rate is low, so informatization has not improved significantly the efficiency of rural tertiary industry integration.

Educational level has a significant positive effect on the integration of rural tertiary industries. The higher the level of education, the higher the financial literacy and the greater the ability to accept and understand digital financial products. At the same time, the higher the level of education, the easier it is for customer financial institutions to use digital inclusive financial resources.

### 5.2. Robustness Test

The benchmark regression results in Table 2 show that the development of digital Pratt & Whitney finance can improve the efficiency of rural tertiary industry integration. In order to make the results more reliable, this paper carries out a series of robustness tests as follows.

#### 5.2.1. Hausman Test

The DID model used in this paper is a fixed-effects panel model. Because there are differences between different provinces, we can use the Hausman test to test the reliability of the fixed-effects model. The Hausman test results are as follows. Because of the *p* = 0 of Model 1 and 2, the fixed-effects model is accepted, and the random-effects model is rejected (Table 3).

#### 5.2.2. The Parallel Trend Test

The double-difference method requires the Experimental and Control Groups to satisfy the parallel trend assumption; that is, before implementing the policy, the Experimental Group and the Control Group have a similar trend of change in the efficiency of rural triple production fusion. Therefore, the Experimental Group and the Control Group were tested by parallel trend, and the parallel trend dynamic effect graph (Figure 1) was drawn by the event study method. As shown in Figure 1, this paper uses the event research method to carry on the regression analysis to the policy implementation of the first 5 years. After the policy implementation of the first 2 years, the regression result shows that in the policy implementation of the first 5 years, the coefficient is not significantly non-zero. This indicates no significant difference between the pilot and non-pilot provinces before implementing the policy and that the parallel trend assumption is satisfied. In addition, after implementing the policy, coefficient is significant and not zero and it further shows that the policy implementation of rural tertiary industry integration efficiency has significantly improved. As shown in Figure 1, the DID model in this paper accords with the parallel trend assumption.

#### 5.2.3. Placebo Test

In order to further reduce the interference of random factors to verify the robustness of the basic conclusions above and to ensure that the development of digital inclusive finance brings about the conclusions of this paper, this paper draws on the ideas of Abadie and Gardeazabal [41]; the Placebo test was performed in randomly assigned provinces. Thirty-one provinces and cities were randomly divided into treatment and control groups and then DID regression was performed according to the random sample. In order to minimize the influence of chance events on the estimation results, a total of 1000 random samples were taken, and 1000 sets of coefficients were obtained by regression according to the benchmark model, as shown in Figure 2. The true policy effect value is abnormal in the placebo test. That is, random factors cannot affect the efficiency of rural tertiary industry integration, and there is no average policy effect in random estimation. It is concluded that the policy effect estimated in this paper is not an inaccurate result caused by uncontrollable factors, and it passes the placebo test.

#### 5.2.4. Counterfactual testing

In addition to the development of digital inclusive finance, there may be other factors in changing the efficiency of rural integration. If the possibility is true, it will cause the conclusion of this paper to be untenable. In order to eliminate this possibility and guarantee the accuracy of the estimation conclusion of the dual difference model, the development of the digital inclusive finance time was tested one year (key_advanve1), two years (key_advanve2), and three years (key_advanve3) ahead of time. The results of the counterfactual test are shown in Table 4.

According to the regression results, the development of Digital Pratt & Whitney Finance has no significant impact on rural tertiary industry integration efficiency. This indicates that without the development of digital inclusive finance, the key, the core explanatory variable, has no significant impact on rural tertiary industry integration efficiency, and there is no systematic error. Therefore, the conclusion of the benchmark model is credible that the development of digital inclusive finance significantly improves the efficiency of rural tertiary industry integration.

## 6. Further Discussion: Heterogeneity Analysis and Conduction Mechanism

### 6.1. Heterogeneity Analysis

The above estimates measure the average effect of digital inclusive finance on rural tertiary industry integration efficiency. They do not reflect the heterogeneous effect of digital inclusive finance on different rural tertiary industry integration levels. In particular, in this paper, we want to study whether digital inclusive finance can promote the integration of rural tertiary industry to a different extent. Are there the same support effects for different rural tertiary industry integration levels? In order to answer these questions, quantile regression is used to estimate the quantile processing effect. The results of quantile DID regression are given in Table 5. From Table 5, it can be seen that the digital inclusive financial policy has a significant promotion effect on the rural integration of the tertiary industry. Still, there is a heterogeneous impact on the different degrees of rural integration of the tertiary industry. The result of coefficient estimation of low percentile is smaller than that of high percentile, which shows that the support effect of developing digital inclusive finance to the provinces with a higher degree of rural tertiary industry integration is more obvious and shows an inverted u shape. In addition, it should also be noted that the change value of the coefficient shows a rising trend with the increase of the percentile, indicating that the supportive effect of China’s digital inclusive financial policy needs to be further improved. It should continue to strengthen the precision support rural integration of the less developed provinces to further improve the efficiency of digital inclusive financial support.

### 6.2. Analysis of Conduction Mechanism

Through the above analysis, we have confirmed that the implementation of the policy can effectively promote the efficiency of rural integration of the tertiary industry. The question is: how does the policy exert its effect? Based on the above theoretical mechanism analysis, this paper takes technological innovation, agricultural modernization level, and risk-sharing mechanism as intermediary variables and constructs an intermediary effect model. In this paper, Key * Tech, Key * Arg, and Key * Insurance, which are the interactive items of the digital inclusive financial index, are added to Formula (6). The regression results are shown in Table 6.

As shown in Table 6, the interactions between development digital inclusive finance and technological innovation, agricultural modernization, and risk diversification were positive and significant at a 1% confidence level. The main reasons are as follows: the development of digital inclusive finance can improve the efficiency of capital allocation, increase the input of rural science and technology research and development, and promote the transformation of agricultural science and technology innovation achievements, thus improving the efficiency of rural integration of the three industries; after the development of digital inclusive finance, the development of agricultural modernization has received more financial support, which has promoted the development of agriculture-related industrial clusters, thus promoting the integration of the three rural industries; moreover, rural integration of the three rural industries, due to the existence of uncertain factors, is often exposed to a variety of risks. With the development of digital inclusive finance, banks, securities, insurance, and other financial institutions, farmers can be provided with corresponding financial products and financial instruments, greatly reducing the risk in integrating the three industries and promoting the rural integration of the tertiary industry.

## 7. Conclusions and Recommendations

This paper takes the development of digital financial inclusion as a quasi-natural experiment. It uses the double-difference method to empirically analyze the impact of the policy on the integration of rural tertiary industries and its transmission mechanism. The results show that (1) the policy is significantly conducive to promoting the integration of the tertiary industries in rural areas. Adjusting the financial ecological environment will help improve the level of integration of the tertiary industries in rural areas. (2) There is a heterogeneous effect. Improving the rural tertiary industry integration level is more significant in the provinces and cities. We should continue to strengthen the precise support of the rural tertiary industry integration in the backward provinces. (3) Verifying its transmission mechanism reveals that the policy promotes the integration of rural tertiary industries by promoting technological innovation, improving agricultural modernization, and building a risk-sharing mechanism. Based on the above research conclusions, this article puts forward the following suggestions:*First, optimize the financial ecological environment and realize the integration of rural tertiary industries.*

A good financial ecological environment is conducive to promoting digital inclusive finance to support the integration of rural tertiary industries. Therefore, the country should continue to improve the financial ecological environment. Given the different characteristics of provinces and cities, differentiated financial eco-environment optimization policies should be formulated to avoid policy uniformity. China has further accelerated the reform of the rural financial system. Rural areas should fully explore local comparative and competitive advantages and accelerate economic development. This will create a good financial ecological environment and give full play to the role of Digital Inclusive Finance in promoting the integration of three rural industries.


*Second, rationally allocate financial resources and accurately support the integration of the tertiary industry.*


As digital financial inclusion has a heterogeneous impact on the integration of tertiary industries in rural areas at different levels, a more reasonable allocation of digital inclusive financial resources is required to support the integration of tertiary industries in rural areas accurately. It is necessary to continue popularizing digital inclusive finance knowledge to farmers and improve the infrastructure construction in rural areas. Furthermore, it is also necessary to continue to promote digital inclusive finance to the countryside, innovate digital inclusive financial products in rural areas, and meet the financial needs of the development of rural tertiary industries. In addition, create diversified financial formats, promote the investment of digital inclusive financial funds into cultivating characteristic industries and competitive industries, and achieve personalized integration of rural tertiary industries.


*Third, improve the transmission mechanism to ensure the development of rural tertiary industries.*


Technological innovation, the level of agricultural modernization, and the risk-sharing mechanism are conducive to promoting the integration of rural tertiary industries. We should vigorously support technological innovation in rural enterprises at the government level through fiscal policies such as tax reduction and exemption. Notwithstanding, we should use fiscal purchases to support agricultural modernization and establish a reasonable rural insurance system, cooperate with the fiscal subsidy mechanism, and promote the integration of insurance and reinsurance. At the level of financial institutions, big data and other technologies are used to provide targeted financial support according to the needs and payment capabilities of rural enterprises and rural operators. Policy-based rural insurance and rural commercial insurance are combined with expanding the scope of insurance and striving to achieve insurance inclusive of the rural tertiary industries to promote the integration of the rural tertiary industries.

There is still room for further improvement in this paper. First, due to the short development time of digital financial inclusion, the period of the digital financial inclusion index used in this paper is only 8 years, and the sample size is relatively small. There may be certain deviations in the empirical regression results. The long-term impact of industrial integration requires further study. Secondly, regarding the concept of the integration of the three industries in rural areas, the theoretical circle has not yet formed a unified view, so there may be deviations in the standard for determining the level of integration of the three industries in rural areas.

## Figures and Tables

**Figure 1 ijerph-19-03363-f001:**
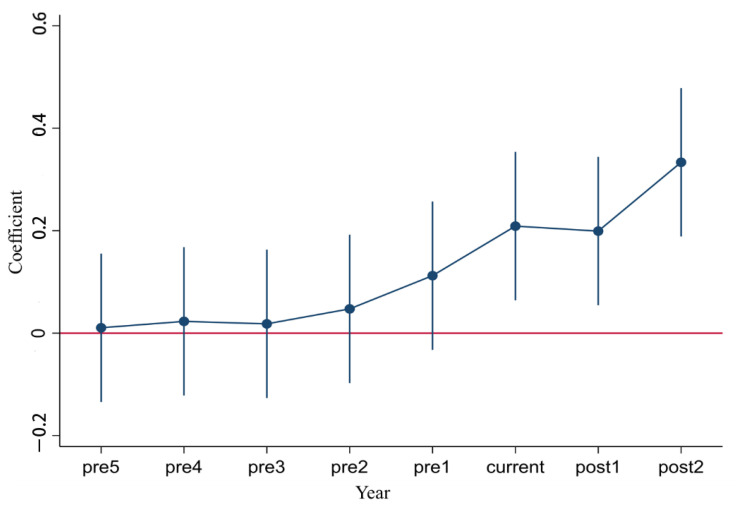
Parallel trend test results.

**Figure 2 ijerph-19-03363-f002:**
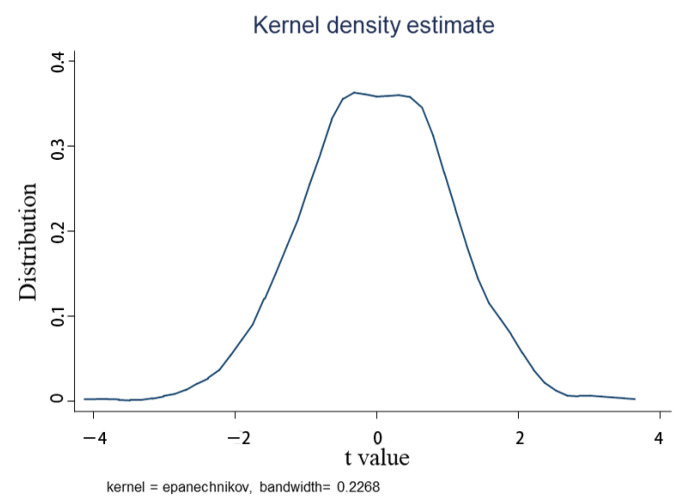
Core density distribution diagram.

**Table 1 ijerph-19-03363-t001:** Descriptive statistics of variables.

Variable	Number of Observations	Mean	Standard Deviation	Minimum	Maximum
Y	248	0.219	0.251	0.005	1
Key	248	0.133	0.34	0	1
Edu	248	0.17	0.06	0.07	0.34
Gs	248	0.249	0.187	0.097	1.291
Fin	248	3.193	1.202	1.518	8.131
Rgdp	248	3.633	2.039	0.692	10.523
Info	248	0.107	0.019	0.067	0.179
Tech	248	9.783	1.627	4.796	13.078
Arg	248	7.207	0.93	3.168	8.752
Insurance	248	5.094	1.105	1.206	7.247

**Table 2 ijerph-19-03363-t002:** Results of baseline regression.

Variables	(1)	(2)
	M1	M2
Key	0.1867768 ***	0.2472517 ***
	(0.0330637)	(0.0422542)
Edu		0.1000542 ***
		(0.0269285)
Gs		0.0827181
		(0.0466475)
Fin		−0.059176 ***
		(0.0117108)
Rgdp		0.0282231 ***
		(0.0046531)
Info		0.1680336
		(0.4590347)
constant	0.1862145 ***	0.2358232 ***
	(0.0135973)	(0.0660355)
N	248	248
R^2^	0.2564	0.2818
City fixed	YES	YES
Year fixed	YES	YES

*** Significant at 1%.

**Table 3 ijerph-19-03363-t003:** Hausman test results.

	Chi-Square Value	*p*-Value
M1	177.140	0.000
M2	288.164	0.000

**Table 4 ijerph-19-03363-t004:** Counterfactual test results.

	(1)	(2)	(1)	(2)	(1)	(2)
	M1	M2	M1	M2	M1	M2
key_advanve1	0.001677	0.003023				
	(0.00800)	(0.00595)				
key_advanve2			0.000569	−0.00182		
			(0.00649)	(0.00563)		
key_advanve3					−0.00171	−0.0043
					(0.00552)	(0.00534)
control	NO	YES	NO	YES	NO	YES
N	248	248	248	248	248	248
R^2^	0.0009	0.0143	0.009	0.121	0.0038	0.0122
City fixed	YES	YES	YES	YES	YES	YES
Year fixed	YES	YES	YES	YES	YES	YES

**Table 5 ijerph-19-03363-t005:** Results of quantile DID regression.

	q10	q20	q30	q40	q50	q60	q70	q80	q90
Key	0.19 ***	0.223 ***	0.236 ***	0.27 ***	0.34 ***	0.45 ***	0.549 ***	0.766 ***	0.616 ***
	(0.009)	(0.024)	(0.03)	(0.028)	(0.033)	(0.037)	(0.055)	(0.0819)	(0.15)
N	248	248	248	248	248	248	248	248	248

*** Significant at 1%.

**Table 6 ijerph-19-03363-t006:** Analysis of conduction mechanism.

	(1)	(2)	(3)
	M1	M2	M3
Key	0.5355956 ***	0.8121402 ***	0.5481415 ***
	(0.1679885)	(0.183524)	(0.1224661)
Key * Tech	0.0662445 ***		
	(0.013545)		
Key * Arg		0.1303135 ***	
		(0.0214283)	
Key * Insurance			0.1224768 ***
			(0.0167246)
control	YES	YES	YES
N	248	248	248
City fixed	YES	YES	YES
Year fixed	YES	YES	YES

*** Significant at 1%.

## Data Availability

Publicly available datasets were analyzed in this study. These data can be found here: http://www.stats.gov.cn (accessed on 19 January 2022).

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
