# Peer review of "Research on Digital Inclusive Finance Promoting the Integration of Rural Three-Industry"

_ijerph, 2022, doi:10.3390/ijerph19063363_

Round 1

Reviewer 1 Report

Authors examine ijerph-1583246 “Research on Digital Inclusive Finance Promoting the Integration of Rural Three Industries Based on Financial Ecological Environment” work seems to be interesting. Authors find that digital financial inclusion is significantly conducive to promoting rural tertiary industry integration and exerts a more significant improvement effect in provinces and cities with higher rural tertiary industry integration levels. Further on the transmission mechanism they find that the policy has promoted the improvement of rural tertiary industry integration efficiency by promoting technological innovation, improving the level of agricultural modernization, and building a risk-sharing mechanism. But following are some the concerns authors can improve on it:

  1. Section 1 need to clearly state what is your aim of the work, rational, major contribution and policy implications.
  2. Section 2 seems to be very weak; I cannot find the relevant literature that talk about the three-transmission channel of rural integration. I advise authors to build strong taxonomy of ‘digital inclusive finance’ with ecological perspective. And clearly show what are your research questions and research hypothesis in section 2 or section 3
  3. In section 4 it is unclear what is the time span of the data, does chosen time span justify your study, why DID, no rational for the control variables (line no. 379)
  4. It’s better to have Hausman specification tests, chose the best model to explain your empirical hypothesis
  5. Authors need to add proper captions at the end of each table

Reviewer 2 Report

Please rewrite the story from a global perspective and add proper citations to government documents. 

Author Response

Response to Reviewer 2 Comments

Point 1: Please rewrite the story from a global perspective and add proper citations to government documents.

Response 1: We have rewritten and mainly expressed:

how to transform, upgrade traditional agriculture, expand and extend the agricultural value chain as the focus of agricultural development in various countries. The integration of the three industries in rural areas is an important focus. In the context of promoting the development of inclusive finance worldwide, more countries have begun to pay attention to the impact of inclusive finance policies on the development of rural industries, and China is a typical example. Therefore, studying the impact of China's digital financial inclusion policy on the integration of rural tertiary industries can provide relevant experience for other countries to learn from. At the same time, provide more policy suggestions for the development of rural tertiary industries in various countries.

We have added some references to government documents.

2021 Central Document No. 1,China Farmers Cooperative, 2021(03):17.

CPC Central Committee and the State Council. Strategic planning for rural revitalization (2018-2022), 2021-06-21.

Literature Research Office of the Central Committee of the Communist Party of China. Selection of Important Documents Since the Nineteenth National Congress of the Communist Party of China, Beijing, Central Literature Research Office, 2019.

The Central People’s Government of the People’s Republic of China. Outline of the 14th five-year plan (2021-2025) for national economic and social development and the long-range objectives through the year 2035,2021-03-11.

Reviewer 3 Report

Dear authors,

Congratulations for your paper.

Although this is not exactly our responsibility, it does not seem that the topic of the article fits perfectly within the Journal scope.

The title should not be too long, as it creates confusion. Try something as “Digital Inclusive Financing to promote the Integration of Rural Three Industries” that could be equally informative but more relevant.

In the abstract the aim is clear, but it can be improved what the study found and how they did it.

In introduction, the research question should be better justified, namely given what is already known about the topic.

References can be strengthened, including recent and relevant references.

The authors present only one reference in the empirical results. This aspect, although not mandatory, can be improved in order to make it possible to understand whether previous studies present concordant results or not.

It could be interesting if authors present the main limitations of the study.

Some suggestions to correction:

  • Specify the meaning of acronyms, e.g. DID, at their first reference.
  • Line 33, “No.1 Central Document in 2021” – please include reference.
  • Line 46, “Strategic Plan for Rural Revitalization (2018-2022)" – please include reference.
  • Line 50, “Fourth Plenary Session of the 19th Central Committee” – please include reference.
  • Line 53, where it reads "my country” replace to “China´s”.
  • Line 56, “14th Five-Year Plan Outline” – please include reference.
  • The three industries studied are explained only in lines 161 and 162. Probably it could result better if they are explained earlier.
  • Lines 176 and 355, the use of capital letter in tow words is not necessary.
  • Lines 355 and 356 the authors wrote “31 provinces and cities in 355 China from 2010 to 2018”; however, in the lines 389 and 391 wrote “31 provinces and cities from 2011 to 2018”. Please clarify the difference between both periods of time.

Round 2

Reviewer 1 Report

Authors have now improved the work